# Application of GC–TOF/MS and GC×GC–TOF/MS to Discriminate Coffee Products in Three States (Bean, Powder, and Brew)

**DOI:** 10.3390/foods12163123

**Published:** 2023-08-20

**Authors:** Xiaolei Fang, Yanping Chen, Jie Gao, Zimu Run, He Chen, Ruoqi Shi, Yingqiu Li, Haihua Zhang, Yuan Liu

**Affiliations:** 1College of Food and Health, Zhejiang Agriculture and Forestry University, Hangzhou 311300, China; xleifang@163.com; 2Department of Food Science & Technology, School of Agriculture & Biology, Shanghai Jiao Tong University, Shanghai 200240, China; catherinechenyp@sjtu.edu.cn (Y.C.); gaojie2020@sjtu.edu.cn (J.G.); tzumu_r@sjtu.edu.cn (Z.R.); chen-he@sjtu.edu.cn (H.C.); shiruoqi@sjtu.edu.cn (R.S.); y_liu@sjtu.edu.cn (Y.L.); 3Secondary College of Cereals and Tourism, Guangxi Vocational College of Technology and Business, Nanning 530005, China; liyingqiu2008@163.com

**Keywords:** coffee, GC×GC–TOF/MS, GC–TOF/MS, volatile organic components

## Abstract

The volatiles in coffee play an important part in the overall flavor profile. In this study, GC–TOF/MS and GC×GC–TOF/MS were used to detect the volatile organic compounds (VOCs) in coffee samples of three different brands at three states (bean, powder, and brew). The differences between the two methods in characterizing VOCs were analyzed using the Venn diagram and PCA (principal component analysis). The important aroma-contributing compounds were further compared and analyzed. The results of the venn diagrams of different coffee samples showed that most VOCs existed in 2–3 kinds of coffee. The PCA of VOCs in different coffee samples showed that the VOCs detected by GC–TOF/MS could distinguish the coffee samples in the different states. GC×GC–TOF/MS was suitable for the further identification and differentiation of the different brands of coffee samples. In addition, pyridine, pyrrole, alcohols, and phenols greatly contributed to distinguishing coffee in three states, and alcohols greatly contributed to distinguishing the three brands of coffee.

## 1. Introduction

Coffee is one of the world’s most consumed beverages. The consumption of coffee and its trade share has been increasingly ascending in China in recent years [1]. The chemical composition of raw coffee beans depends on many factors, such as varieties, origins, and planting conditions (soil, temperature, water, etc.) [2]. The sensory profile is a very important indicator of coffee quality, which can be evaluated based on the international coffee flavor wheel. The wheel was developed by the American Specialty Coffee Association and divided into two parts including tastes and aromas [3]. Coffee drinks present a certain sour and bitter taste, of which the sour taste is mainly caused by organic acids such as citric acid and malic acid, and the bitter taste mainly comes from alkaloids, chlorogenic acid baking products, and Maillard reaction products. The aroma of coffee varied significantly among the samples. Raw coffee beans are green and beany, but roasted coffee beans appear brown or tan in color, with consumers’ preferring roasted, nutty, floral, chocolate, etc. So far, more than 1000 volatile compounds have been found in coffee, such as pyrazines, furans, ketones, aldehydes, etc., and various volatile compounds work together to produce the coffee-specific aroma [4].

The coffee flavor is affected by the processing technology (baking temperature and time), storage conditions, grinding and brewing process, etc. [5,6,7]. The typical attractive flavor of roasted coffee beans is mainly generated by the Maillard reaction, Strecker degradation, caramelization, and pyrolysis reactions during the high-temperature roasting process [8,9]. The preparation conditions include the grinding particle size of the roasted coffee beans and the temperature of the hot water poured over the ground coffee, which also affect the level of extracted coffee flavor components [10].

Some sulfur-containing compounds, such as 2-furfurylthiol, methanethiol, and 2-methyl-3-furanthiol, as well as furans and pyrazines, are among the most significant volatiles in terms of coffee flavor, and extremely influence the sensory profile of coffee [11,12,13]. The sulfur-containing compounds, furans, and pyrazines are at low concentration levels in coffee, but their extremely low odor thresholds mean that they have a great olfactory impact. Gas chromatography–ion mobility spectrometry (GC–IMS) and electronic nose (E-nose) has been used to identify and distinguish volatile compounds in coffee from different origins through principal component analysis (PCA) [14]. However, the traditional one-dimensional GC (1D-GC) has defects of poor separation, insufficient peak capacity, and serious co-current interference. In recent years, two-dimensional gas chromatography–time-of-flight mass spectrometry (GC×GC–TOF/MS) has been widely used due to its stronger peak separation ability, especially in separating numerous subtle volatiles in complex sample matrices [15]. It has better shape quality, higher peak purity, and increased accuracy of qualitative and quantitative data [15]. The chromatographic peaks are gathered in a two-dimensional (2D) plane by two columns, so each volatile has two retention indexes with a stable position. Since GC×GC–TOF/MS can analyze thousands of components in a short time due to its extremely high spectral generation rate (up to 500 spectra/s), it has been applied to study the flavor of Baijiu [16], Chinese dry-cured hams [17], and brewed coffee [18,19].

Therefore, the objectives of the study were (1) to characterize the volatile organic compounds (VOCs) of three commercially available coffee products in three states (bean, powder, and brew); (2) to compare the VOCs produced in complex coffee matrix by GC×GC–TOF/MS method and GC–TOF/MS method, and (3) to investigate the subtle but important aroma contributing compounds, such as sulfur-containing compounds, furans, pyrazines in varied coffee samples.

## 2. Materials and Methods

### 2.1. Coffee Samples

Roasted coffee beans (moderately roasted) were bought at Shanghai retail coffee store and stored in vacuum-sealed bags. Three Arabica coffee bean brands were purchased and abbreviated as ST, CO, and PA. The coffee samples were prepared in three states, including beans, ground powder, and brews. Each sample was prepared on the day of the experiment following our previous experiments [14]. No treatment was performed on the beans. The coffee powder was processed by grinding the coffee beans using a manual coffee grinder. The powder passing through a 700 μm sieve was collected and immediately stored in odorless, sealed aluminum bags. The coffee brew was prepared by placing the ground coffee powder (5.5 g) and 100 mL of 92 °C hot water into the coffee pressure pot, squeezing three times, packing 4 mL into 20 mL headspace vials, and sealing immediately.

### 2.2. Headspace Solid-Phase Microextraction

The coffee samples were prepared for headspace solid-phase microextraction (HS-SPME) according to three states (beans, powder, and brew). Compared with other extraction methods, solid-phase microextraction can save 70% of sample pretreatment time and meet the needs of the rapid detection of large quantities of coffee samples in a shorter time [20,21]. The coffee bean and coffee powder weights were 1.5 g for GC–TOF/MS and 0.75 g for GC×GC–TOF/MS. The coffee brew was measured to a volume of 4 mL for GC–TOF/MS and 2 mL for GC×GC–TOF/MS. The samples, according to the brand and state, were immediately placed and sealed into 20 mL headspace vials. The internal standard (IS) was prepared by dissolving 2-methy-3-heptanone (purity 95%, GC standard, liquid, Mobile Biotechnology Co., Ltd., Shanghai, China) in methyl alcohol (purity ≥ 99.9%, GC standard, liquid, Aladdin, Shanghai, China) to the final concentration of 0.41 mg/mL. The IS solution (1 μL) was then manually added to vials using a 10 μL syringe before the SPME process [22].

Volatile compounds in the prepared samples were extracted using the 50/30 μm DVB/CAR/PDMS fiber (Stableflex, Supelco, Bellefonte, PA, USA) [9,23]. The samples were equilibrated at 40 °C for 10 min. The volatiles were extracted at 40 °C for 20 min and subjected to desorption for 5 min [12].

### 2.3. GC–TOF/MS Condition

The experiment was conducted at the Instrument Analysis Center, Shanghai Jiao Tong University. A DB-Wax GC column (length 30 m, inner diameter 0.25 mm, film thickness 0.25 μm; Agilent 7890 B) coupled with a mass spectrometer (Pegasus BT, MI, United States) was used for the separation of volatile components. The column temperature was held initially at 60 °C for 5 min, increased to 180 °C at the rate of 3 °C/min, then raised to 250 °C at the rate of 10 °C/min and held for 5 min [12]. Helium was used as the carrier gas with a 1 mL/min flow rate. The mass spectrometer conditions were set as follows: ion source and interface temperatures were 300 °C and 275 °C, respectively. The ionization potential of MS was 70 eV, and the scan range was 50 to 350 *m*/*z* for 2 s [24].

The retention times of n-alkanes (C_4_–C_30_; Sigma, Aldrich Trading Co., Ltd., Shanghai, China) were obtained at the same experimental conditions as mentioned above in order to calculate the retention index (RI) of the volatile organic compounds (VOCs).

### 2.4. GC×GC–TOF/MS Condition

The GC×GC–TOF/MS analyses were performed using a LECO Pegasus 4D instrument (LECO Corp., St. Joseph, MI, USA) equipped with a non-moving quad-jet cryomodulator. DB-WAX (30 m × 250 μm × 0.25 μm) and DB-17HT (1.9 m × 100 μm × 0.10 μm) were used as the first and the second-dimensional columns, respectively. Helium (99.999%) was used as the carrier gas at a flow rate of 1 mL/min and the injection volume was 1 μL. The initial temperature of the first column was held for 5 min at 60 °C. After that, the temperature was increased to 240 °C at 3 °C/min and then held for 5 min. The initial temperature of the second column was held for 5 min at 65 °C. After that, the temperature was increased to 245 °C at 3 °C/min and then held for 5 min. The modulator was offset in relation to the secondary oven, the modulation cycle was 4.0 s, and the last was 0.7 s.

The mass spectrometry conditions were as follows. Electron ionizing source: ionizing energy, 70 eV; the mass scanning range was 41 to 415 u. The temperature of the conversion line was 250 °C, and the ion source temperature was 200 °C.

Data were processed and consecutively visualized on 2D and 3D chromatograms using LECO ChromaTOF™ software. The retention index (RI) was calculated using alkane standards (C_4_–C_40_; Sigma, Aldrich Trading Co., Ltd., Shanghai, China) at the same experimental conditions as mentioned above.

### 2.5. Identification of Volatile Components

The tentative identification of VOCs for GC–TOF/MS was based on mass spectra and calculated RI values. The MS and RI data were compared with those reported in the database (NIST). VOCs from GC×GC–TOF/MS data were tentatively identified after comparing their linear retention indices (RIs), first- and second-dimension retention times, and mass spectra with the NIST 17 database. Semi-quantification was achieved by relating the GC peak area of each VOC to the peak area of the IS (3-heptanone). The relative quality of each VOC was calculated based on Equation (1) below.
m _VOC_ = Area _VOC_ × m _IS_/Area _IS_(1)

All analyses were carried out in triplicate; the results are shown as mean ± standard deviation (SD). Original statistical data were analyzed by SPSS 20.0, and chart processing was processed by Origin 2018. The Venn diagrams were graphed by jvenn, a JQuery plugin based on J.C. Oliveros’ venny tool, which is an interactive online tool for comparing lists with Venn diagrams.

## 3. Results

### 3.1. Identification of VOCs in the Coffee Bean, Powder, and Brew

Table 1 shows the number of VOCs identified in the coffee bean, powder, and brew using GC–TOF/MS (G) and GC×GC–TOF/MS (GG), respectively. The VOCs were divided into 13 groups, including hydrocarbons, aldehydes, ketones, acids, phenols, alcohols, esters, pyridines, pyrroles, pyrazines, furans, sulfur-containing compounds, and other compounds. The total number of VOCs identified by G was less than 100, but the total number of VOCs identified by GG ranged from 175 to 374. The results were the same as the previous investigation, in which GG could identify more VOCs in the food samples [17]. Figure 1 shows an example of the chromatogram results of ST coffee beans separated using GC–TOF/MS (Figure 1A) and GC×GC–TOF/MS (Figure 1B,C). The peak number of GC×GC–TOF/MS was higher than that of GC–TOF/MS (Figure 1A,B). The chromatographic peaks of GC×GC–TOF/MS were gathered in a two-dimensional (2D) plane by two columns, so each volatile had two retention indexes with a stable position (Figure 1C).

The ketone is the top VOC group in CO, PA, and ST in bean, powder, and brew states. Especially, GG identified 88, 95, and 95 VOCs in the three brands of coffee powder. In addition, the numbers of identified aldehydes and ketones in beans, powder, and brews using GG were 4–6-fold greater than those using G. The oxygenated compounds, such as aldehydes and ketones, are generated during high-temperature roasting, when proteins and carbohydrates undergo the Maillard reaction, Strecker degradation, caramelization and pyrolysis [25]. The number of furans identified by G ranked second. Furans can be generated by the heat treatment of Maillard reaction precursors or lipids [26]. Glucose and alanine were important precursors of furan, of which the formation largely relied on the heating temperature [27]. The number of furans identified by GG was almost twice that by G.

Amino acids provide aminoketones that interact with aldehydes, and the Strecker reaction to generate pyrazine yields a series of volatile heterocyclic compounds with aromatic activity, such as pyrazine, pyrrole, and pyridine. Sulfur-containing compounds are the most important class of volatile aromatic compounds, originating from the thermal degradation of sulfur-containing amino acids in the presence of sugars [28]. The number of pyridines and pyrroles identified using G was only 0–3, but the number identified by GG ranged from 2 to 10. Pyrazine and sulfur-containing compounds are also important contributors to coffee flavor. Only 5–10 pyrazines and less than 5 sulfur-containing compounds were identified using G. However, 10–31 pyrazines and sulfur-containing compounds were identified using GG.

The number of acids and phenols identified by G and GG was almost the same. Though acids and phenols were not plentiful in the coffee samples, they could have contributed to the sour smell [29]. In this investigation, GG identified three-fold more hydrocarbons and other compounds than G. These two groups were not important to the overall odor profiles of the coffee samples due to their high thresholds.

After grinding, the number of aldehydes, ketones, alcohols, and sulfur-containing compounds increased. The number of VOCs identified in coffee brews decreased compared to coffee beans and powder. After the grinding process, more VOCs are released from the bean matrix, resulting in the most abundant VOC varieties [12]. The coffee brews prepared by hot water were better for extracting water-soluble VOCs, such as polar compounds [2]. The reduced number of VOCs in coffee brews might be due to the loss of lipid-soluble VOCs.

Venn diagrams were further used to reveal the VOCs identified in the three brands of coffee (CO, PA, ST) under three treatments using G and GG (Figure 1). Figure 2A–C represents the number of VOCs identified in the coffee beans, powder, and brews, respectively. In Figure 2A, the number VOCs identified using G ranged from 33–43 in the three brands of coffee beans, but the number of VOCs identified using GG ranged from 66–135. Only 6 VOCs overlapped in all the samples, 18 VOCs were shared by five samples, and 8 VOCs were shared by four samples. The results suggested that the VOCs contained in the coffee beans varied differently from each other. This was the same for the VOCs identified in the coffee powder and brews. For the coffee powder in Figure 2B, 16 VOCs were shared by six samples. For the coffee brews in Figure 2C, 15 VOCs were shared by six samples. In terms of the VOCs, most of them were present in at least two of the three species of coffee, and only a small number of VOCs were specific to a particular kind of coffee. There were 39 VOCs that were only identified in one type of coffee bean. In addition, 32 VOCs were not shared among the three brands of coffee powder, and 35 VOCs were not shared among the three brands of coffee brews.

### 3.2. PCA Based on VOCs Identified by GC–TOF/MS and GC×GC–TOF/MS

The VOCs identified using G and GG were analyzed by PCA (Figure 3). Figure 3A shows the PCA of the VOCs identified by G. The first principal component (PC1) accounted for 31.3% variance, and the second principal component (PC2) accounted for 17.0% variance. The cumulative contribution of PC1 and PC2 was 48.3%. The three coffee states were distributed in different spatial locations. The coffee beans were all located on the negative side of the *y*-axis, the coffee powder was located in the first quadrant, and the coffee brew was located on the negative side of the *x*-axis and was distributed in the second quadrant. The different brands of coffee were relatively separated. The distance between PA and CO was closer, and PA was relatively far away from them in the coffee beans. For the coffee powder, all three brands were gathered together. On the contrary, the three brands of coffee were relatively separated, especially ST. The above results indicated that different coffee samples could be clearly distinguished according to the state rather than the brand. The VOCs of the coffee samples in this experiment mainly depended on the treatment rather than the brand. On the other hand, the different brands of coffee could only be distinguished in the brewing stage.

Figure 3B shows the PCA of the VOCs identified using GG. PC1 accounted for 32.8% variance, the second principal component (PC2) accounted for 23.0% variance, and the cumulative rate of the first two PCs was 55.8%. The different coffee states were still distributed in different spatial locations, but they did not have as clear boundaries as in the PCA of G. The coffee beans were also located on the negative side of the *y*-axis, the coffee powder was located in the second quadrant, and the brewed coffee samples were mainly distributed along the negative *y*-axis; only ST was distributed in the first quadrant. Compared with the results of G, ST was more obviously separated from the other brands, and the PA and CO brands of coffee were more aggregated in the coffee brews. In the coffee powder, the coffee samples of different brands showed a better separation effect. In the coffee beans, the coffee brand ST separated from the group, which was the same for PA in the results of G. The results of GG showed that the separation of the different brands of coffee was more obvious, especially in the two states of powder and brew. In other words, GG was more conducive to the distinction between some brands in coffee powder and coffee brews.

### 3.3. Identification of Key VOCs in Coffee Samples

The VOCs of roasted beans, ground powder and brewed coffee were studied and divided into different groups. Previous reports have shown that some VOCs, though not high in concentration, are very important due to their low thresholds [30]. These compounds are summarized and listed in Appendix A, including the pyridine, pyrrole, pyrazine, alcohol, phenol, furan, and sulfur-containing compounds. Appendix A lists those VOCs identified using G, and Appendix A lists those VOCs identified using GG.

The pyridines, pyrroles, and pyrazines, also known as nitrogen-containing heterocyclic compounds, are produced from the Maillard reaction and impart the characteristic roasted or toasted flavors [31]. The concentrations of pyridines and pyrroles were relatively lower than pyrazines in the coffee samples (Appendix A). The concentrations of pyridine were relatively high in the three brands of coffee in the bean and powder state, ranging from 1.1982 to 4.1309 (Appendix A). Pyridine is known to have bitter, burnt, roasted, and astringent characteristics, produced by trigonelline degradation and Maillard reactions [32]. It is especially present in coffee that is subjected to high temperatures or strong roasting processes [24]. Pyrrole was only detected in the coffee beans, and the concentrations ranged from 0.0420 to 0.1319 (Appendix A). Pyrroles are usually related to nutty, hay-like, and herb aromas. There were three pyrroles that were identified using G, and most of them were only present in beans. Only one compound, 1-(2-furan methyl)-1H-pyrrole, with hazelnut and coffee aromas [33], was detected in all samples. The other two pyrroles, 1-methyl-1H-pyrrole and pyrrole, were only present in the coffee beans. In a study on the flavor components and quality characteristics of espresso, 1-(2-furan methyl)-1H-pyrrole was also classified as green and had a vegetable aroma [34]. There were eight pyrroles detected using GG, two of which, 2-ethyl-4-methyl-1H-pyrrole and 3-methyl-1H-pyrrole, were only present in the coffee powder. Only one compound, 1-butyl-1H-pyrrole, was detected in the brewed coffee. This result corresponds to the PCA results showing the powder and brew samples located separately on the map.

Pyrazines are known to have roasted and earthy characteristics in coffee. Pyrazines can mask the odor of other compounds, such as the butter flavor associated with 2,3-butanedione and 2,3-pentanedione levels [35]. There were 23 more pyrazines that were identified using GG than using G. From the list of VOCs in Appendix A, the concentrations of methyl pyrazine, 2,3-dimethyl pyrazine, 2,5-dimethyl pyrazine, and 2,6-dimethyl pyrazine were the top four pyrazines that were detected in almost all the samples. Methylpyrazine has nutty, toasted, and chocolate aromas [36]. It is formed during the roasting process as a result of the Maillard reaction between sugar and amino acids [37]. One study showed that the formation of pyrazine compounds was affected by the reaction temperature, reaction time, and reactant concentration in glucose and L-alanyl-L-glutamine (Ala-Gln-ARP) reaction systems [38]. Of the remaining three pyrazines, 2,3-dimethyl pyrazine had roasted, nutty, and caramel flavors [39,40], 2,5-dimethyl pyrazine had nutty and peanut flavors [41,42], and 2,6-dimethyl pyrazine had sweet, fried, and nutty flavors [36]. For all the identified pyrazines, sucrose is the main carbohydrate precursor. The side products of the Strecker degradation of amino acids can release aroma-active volatile aldehydes, which are crucial for the flavor of coffee [43]. In addition, the fragmentation of the side chains may lead to the formation of pyrazines. Since the pyrazine generation reactions consume free sugars, it has been suggested that they are in competition with other reaction pathways consuming intact sugar skeletons during coffee roasting, such as the formation of 4-hydroxy-2,5-dimethyl-3(2H)-furanone [44].

Alcohols and phenols are also very important VOCs that can contribute to the smoke, sweet, and medicinal characteristics, or evoke a burning sensation associated with spice and clove flavors. These compounds also contribute to the typical flavor of dry-cured ham [17]. Fat oxidation and the Strecker reaction is the main method for producing alcohols [45]. There were 50 more alcohols that were identified using GG than using G. The type and content of alcohols showed a decreasing trend for the powder, bean, and brew, except for 2-furan methanol with its caramel and smoky aroma [30], which was only detected in brews using GG. In addition, among the three brands, the content of 2-furan methanol in CO was the highest. However, this result is inconsistent with a study on Yunnan coffee, in which the substance was detected in all three states (bean, powder, and brew) [30]. The differences may be due to the varied raw coffee materials used in the two studies. In addition, 2-furan methanol acetate with its fruity flavor was abundant in the coffee powder of the CO brand, which may be one of the reasons for the relatively separated location of the CO brand in the powder state on the PCA diagram of GG (Figure 3B). The type and number of phenols were basically the same, but the contents of maltol, phenol, and 2-methoxy-phenol in these phenols were relatively high. Maltol, with its caramel smell, can be formed from the degradation of maltose and Amadori intermediates. Maltol can be used as a marker to determine the roasting conditions of coffee and to ensure flavor quality [39]. It was at a higher level in the beans and powder, so could be detected using G and GG, but it was at a trace level in the brew state, so could only be detected using GG. A study of common roasting defects in coffee showed that most aroma compounds, especially maltol, increased when coffee beans were roasted at higher initial temperatures and longer roasting times [46]. Phenol has a smoky and sweet flavor that is present in all brands and states of coffee [30]. 2-methoxy-phenol has a meaty odor, and its content is higher in coffee powder, especially when detected using GG. In the analysis of odor-active compounds in cereal coffee, 2-methoxy-phenol was also detected [47].

Furans are nowadays receiving special attention in coffee samples since they have previously been the main chemical class found in ground Arabica coffee and all ground and espresso coffee samples [48]. Usually, they are related to the flavor of foods and beverages, providing pleasant characteristics. The furan concentration in coffee drinks depends on its content in coffee powder, as well as on the brewing procedure. The levels of furans in brewed coffee are more abundant than in other treatments [12]. Furan derivatives, such as methyl, aldehyde, alcohol, or ester analogs, are of great concern because of the formation of possible reactive metabolites. The particle size and variety of coffee are closely related to the content of furan [49]. Furfural, together with the corresponding alcohol (furfuryl alcohol), occurs in many fruits, tea, coffee, and cocoa. The furan derivatives detected in coffee samples are mainly alkyl substituents, of which 2,5-dimethylfuran has an ethereal aroma and 2,3,5-trimethylfuran has a coffee aroma [36]. Dimethyl trisulfide, with its cabbage odor [39], is a sulfur-containing compound and might be produced from methionine, cysteine, and ribose [50].

## 4. Conclusions

In this investigation, GC–TOF/MS and GC×GC–TOF/MS were used to analyze the VOCs of three brands of coffee under three conditions (bean, powder and brew). The total number of VOCs identified by G was less than 100, but the total number of VOCs identified by GG was between 175 and 374. The main types of different compounds were ketones, alcohols, esters, hydrocarbons, aldehydes, sulfur-containing compounds, furans, and pyrazines. After grinding, the number of aldehydes, ketones, alcohols, and sulfur compounds increased. After brewing, the number of VOCs in coffee decreased. The Venn diagram of VOCs showed that the common compounds in the beans, powder, and brews were only 6, 16, and 15, respectively, and the VOCs were different among the three states. The PCA results of the VOCs identified by G clearly showed that different coffee samples could be clearly distinguished by state rather than brand. Compared with the results of G, GG could detect more VOCs, thus promoting the further separation of different brands of coffee. Generally, pyridine, pyrrole, alcohols, and phenols greatly contributed to the distinction of the different coffee states. These compounds were only present in one or two different states or with significant differences in content, such as 1-(2-furan methyl)-1H pyrrole with its hazelnut and coffee aroma, 2-furan methanol with its caramel and tobacco flavor, and 2-methoxy phenol with its meat flavor. Additionally, alcohols greatly contributed to the distinction among the three brands of coffee, such as 2-furan methanol acetate with its fruity flavor. The results of GG showed that the separation of the different brands of coffee was more obvious, especially in the two states of powder and brew. In other words, GG was more conducive to the distinction between some brands in terms of coffee powder and coffee brew in a short time. In conclusion, compared with GC–TOF/MS, GC×GC–TOF/MS with its better shape quality and higher peak purity increased the accuracy of the qualitative and quantitative data, which could efficiently detect more VOCs, including those at trace level that played a role in coffee flavor. The technology is suitable for the further identification and differentiation of the different brands of coffee samples.

## Figures and Tables

**Figure 1 foods-12-03123-f001:**
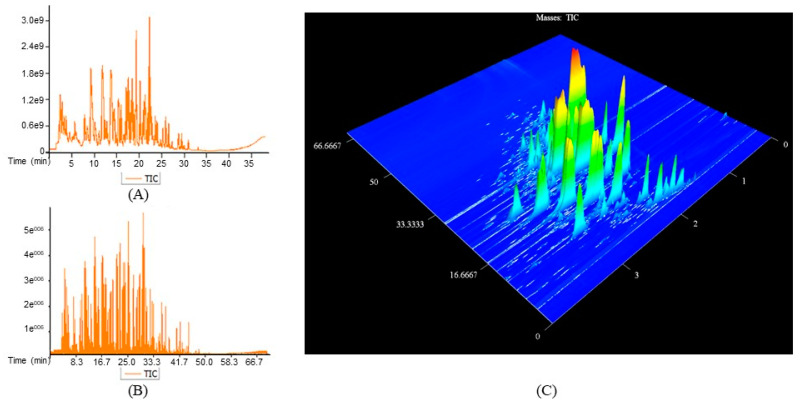
GC–TOF/MS and GC×GC–TOF/MS chromatograms of ST coffee beans. (**A**) Total ion flow chromatograms of ST from GC–TOF/MS. (**B**) Total ion flow chromatograms of ST from GC×GC–TOF/MS. (**C**) 3D image of ST from GC×GC–TOF/MS.

**Figure 2 foods-12-03123-f002:**
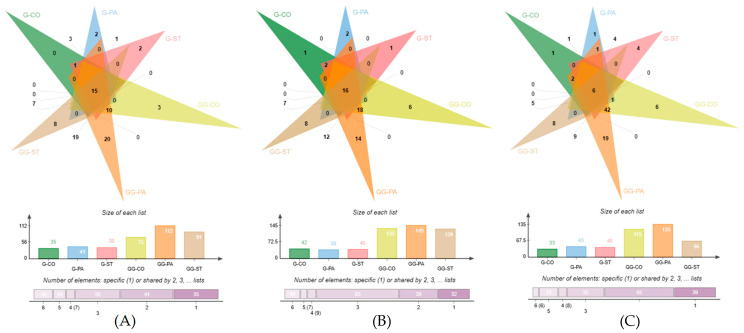
Venn diagrams of VOCs identified in three brands of coffee (CO, PA, ST) under three treatments. (**A**–**C**) Coffee bean, powder, and brew, respectively; G represents VOCs identified by GC–TOF/MS; GG represents VOCs identified by GC×GC–TOF/MS. The bottom of the column bar chart shows the number of VOCs identified in each coffee brand using G or GG. The stacked bar chart shows the number of VOCs shared among different brands that were identified using G or GG.

**Figure 3 foods-12-03123-f003:**
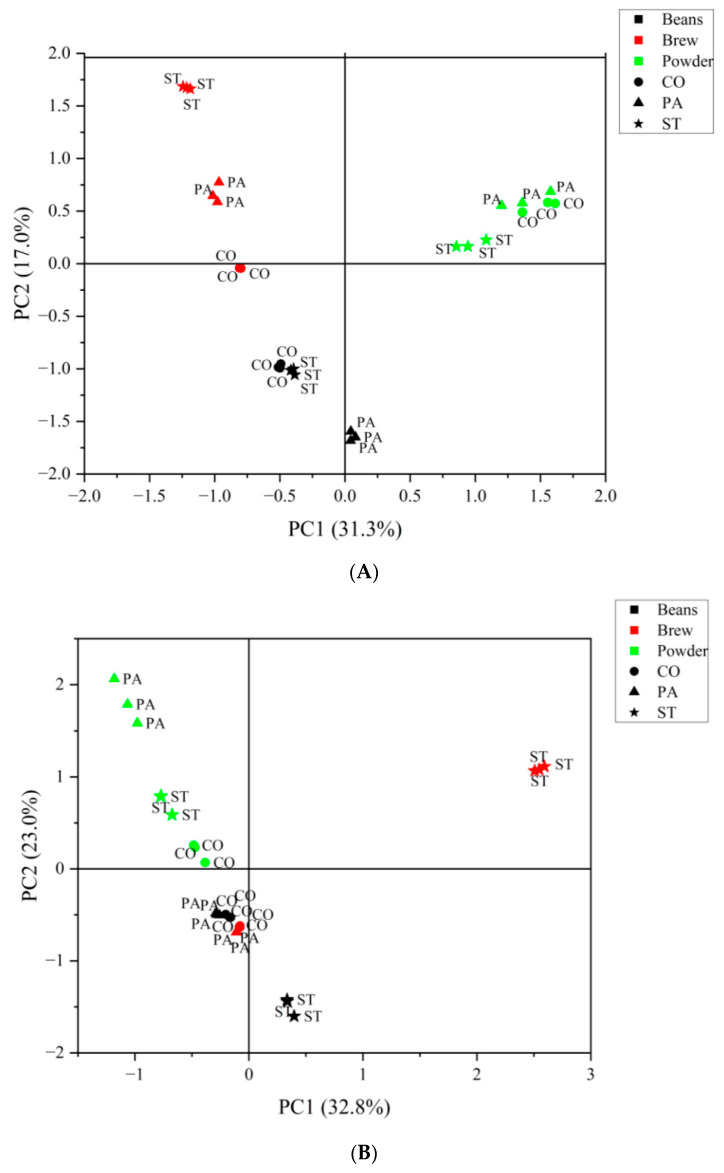
Product space based on PCA of VOCs identified from three brands of coffee (CO, PA, ST) under three treatments (bean, powder, and brew) using GC–TOF/MS (**A**) and GC×GC–TOF/MS (**B**). The black color represents bean, the red color represents brew, and the green color represents powder. Different shapes represent the brands: the circle for CO, the triangle for PA, and the star for ST.

**Table 1 foods-12-03123-t001:** The number of VOCs identified in three brands of coffee in different states (beans, powder, and brews) using GC–TOF/MS and GC×GC–TOF/MS, respectively. The colors correspond to the coffee brands, light yellow represents CO, white represents PA, and light blue represents ST.

	GC–TOF/MS									GC×GC–TOF/MS								
	Beans			Powder			Brew			Beans			Powder			Brew		
	CO	PA	ST	CO	PA	ST	CO	PA	ST	CO	PA	ST	CO	PA	ST	CO	PA	ST
Hydrocarbons	6	8	8	6	6	4	8	5	13	21	28	23	28	29	39	13	31	33
Aldehydes	4	5	3	5	8	7	10	10	10	21	25	13	26	28	26	17	28	18
Ketones	13	14	15	14	13	15	8	7	10	69	83	41	88	95	95	51	63	50
Acids	3	5	10	10	6	7	9	6	8	13	13	5	13	14	16	10	7	6
Phenols	5	7	7	8	8	8	8	10	9	11	10	5	11	8	10	6	10	7
Alcohols	4	9	8	8	6	7	5	6	5	26	36	19	32	40	30	19	25	19
Esters	8	8	7	5	5	6	1	3	7	25	37	15	31	41	36	14	25	19
Pyridines	2	2	2	2	2	1	1	0	0	7	8	3	7	10	6	2	5	4
Pyrroles	2	3	3	1	1	1	1	1	1	4	4	2	7	7	5	5	5	5
Pyrazines	8	9	6	8	8	8	6	5	7	27	31	10	25	23	28	14	17	16
Furans	9	10	10	11	11	11	11	14	12	20	23	13	27	28	26	15	26	25
Sulfur-containing compounds	3	3	4	4	2	4	3	5	4	19	22	13	23	29	24	12	24	15
Other compounds	6	6	6	6	5	10	2	5	8	24	24	13	24	32	33	8	20	12
Total	73	89	89	88	81	89	73	77	94	285	345	175	342	384	374	186	286	229

## Data Availability

The data presented in this study are available on reasonable request from the corresponding author.

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
