# Peer review of "Application of GC–TOF/MS and GC×GC–TOF/MS to Discriminate Coffee Products in Three States (Bean, Powder, and Brew)"

_foods, 2023, doi:10.3390/foods12163123_

Round 1

Reviewer 1 Report

Application of GC-TOF/MS and GC×GC/TOF-MS to discriminate coffee products at three statuses (bean, powder, and brews)

The subject of the work is very interesting. The research problem was correctly formulated and implemented. Appropriate research methods were selected. The determinations were carried out in an appropriate manner and conclusions were drawn.

The abstract is written correctly.

In the introduction, the problem was properly outlined and the aim of the research was presented. The methodology contains all the necessary information.

The results were presented in an unambiguous and understandable way. A detailed statistical analysis deserves attention.

Two small notes to work with:

1. The conclusions should be clearer and better highlight the differences between the methods studied and the benefits of using a better method. Please edit them.

2. The references both in the review and in the discussion of the results should be more recent. Please enter some newer items that are available in large quantity.

Author Response

Dear reviewer,

Best regards,

Xiaolei Fang

Reviewer 2 Report

The presented work is of significant scientific interest, mainly highlighting the results obtained from the comparison of the addressed analytical techniques. These identified differences allow for better outcomes in more focused studies aimed at understanding the aroma of coffee, a matrix of great importance both scientifically and socially. 

Methodology:

It would be beneficial for the authors to provide a more robust justification for why they chose the extraction conditions using SPME (Solid Phase Microextraction). While references are cited, they do not fully define the conditions employed.

A clear explanation is needed regarding how the quantities of coffee were selected for each type of sample, as there is a significant difference between the amount of coffee used for solid and powdered samples, which could significantly influence the results obtained.

Results:

It is observed that tables S1 and S2 lack units, and the title does not fully correspond to the presented information. It is important to correct this to make the data more comprehensible and precise.

I consider it necessary for the authors to include the chromatograms obtained from both techniques, as it would provide a better visual understanding of the results

Author Response

(The authors gave the same response as above.)
